

# The switch from one substance-of-abuse to another: illicit drug substitution behaviors in a sample of high-risk drug users

Barak Shapira[1,2], Paola Rosca[3], Ronny Berkovitz[2], Igor Gorjaltsan[4] and Yehuda Neumark[1]

[1] Braun School of Public Health and Community Medicine, Hebrew University of Jerusalem, Jerusalem, Israel
[2] Division of Enforcement and Inspection, Israel Ministry of Health, Jerusalem, Israel
[3] Department for the Treatment of Substance Abuse, Israel Ministry of Health, Jerusalem, Israel
[4] Beer-Sheva Mental Health Center, Beer-Sheva, Israel

Corresponding author
Barak Shapira,
barak.shapira@mail.huji.ac.il,
shapib@gmail.com

## ABSTRACT

**Background**. Substitution can be defined as the consciously motivated choice to use one drug, either licit or illicit, instead of another, due to perceptions of cost, availability, safety, legality, substance characteristics, and substance attributions. Substitution represents a potential risk to drug users, mainly when substitutes are of higher potency and toxicity. This study offers a basic conceptualization of illicit substitution behavior and describes substitution patterns among users of two highly prevalent drugs of abuse—heroin and cannabis.

**Methods**. Here, 592 high-risk drug users undergoing pharmacological and psycho-social treatment were interviewed. Patients were asked questions about current drug use, lifetime substitution, and substitution patterns. Descriptive statistics, chi-square tests of independence, and multinomial logistic regressions were used to identify and test correlates of substitution patterns for heroin and cannabis.

**Results**. Of the 592 drug users interviewed, 448 subjects (75.7%) reported having substituted their preferred drug for another illicit substance. Interviews yielded a total of 275 substitution events reported by users of cannabis, and 351 substitution events reported by users of heroin. The most frequently reported substitution substances for responders who preferred heroin were illicit non-prescribed ''street'' methadone (35.9%), followed by oral and transdermal prescription opioids (17.7%). For responders who preferred cannabis, substitution for synthetic cannabinoid receptor agonists (33.5%) followed by alcohol (16.0%) were the most commonly reported. Age at onset–of–use ($p < 0.005$), population group ($p = 0.008$), and attending treatment for the first time ($p = 0.026$) were significantly associated with reported lifetime substitution. Past-year use of stimulants, heroin, hallucinogens, methylenedioxymethamphetamine (MDMA), and novel psychoactive substances were—at the 95% confidence level—also significantly associated with reported lifetime substitution. In multivariate analysis, the odds for methadone substitution among heroin users were significantly affected by age at onset-of-use, type of treatment center, and education. Odds for substitution for synthetic cannabinoid receptor agonists among cannabis users were significantly affected by age, population group, type of treatment center, and education.

**Conclusion**. Self-substitution behavior should be considered by clinicians and policy-makers as a common practice among most drugusers. Substitution for street methadone provides evidence for the ongoing diversion of this substance from Opioid Maintenance Treatment Centers, while the prominence of substitution of synthetic cannabinoids among dual-diagnosis patients should be regarded as an ongoing risk to patients that needs to be addressed by clinicians. Analysis of additional substitution patterns should provide further valuable insights into the behavior of drugusers.

## INTRODUCTION

Drug substitution (also referred to as drug displacement or replacement) is, in essence, a druguser's conscious switch from one drug to another. Substitutions may be motivated by several factors including the drug user's perception of higher drug purity, greater availability or lower cost of the substitute drug, or positive expectations in general regarding the effects of the substitute drug (*Moore et al., 2013*). The concept of drug substitution is most commonly associated with opioid maintenance treatment (OMT) whereby a person who is dependent on opioids (e.g., heroin) receives a prescribed drug, typically methadone, buprenorphine, or codeine to minimize opioid-related harm (*Maddux & Desmond, 1992*; *Backmund et al., 2001*). OMT is a widely accepted harm-reduction strategy that is medically viable, relatively safe, and effective (*Dreifuss et al., 2013*;*Orman and Keating, 2009*). Conversely, non-medically sanctioned substitution—the focus of this study—can be conceptualized as a pattern of drug switching performed outside of the formal treatment setting and initiated by the druguser.

User-initiated substitution of heroin and cocaine was documented as early as the first half of the 20th century (*Warburton, 1992*), and has since become a common feature of drug use with regular appearance of newer synthetic drugs that displace or replace older established drugs (*Henderson, 1988*). In the past two decades, in particular, the phenomenon of substitution has gained interest due to the advent of novel psychoactive substances (NPS) that are often depicted as being "substitution drugs" of established substances (*Coppola & Mondola, 2012a*; *Coppola & Mondola, 2012b*; *Hillebrand, Olszewski & Sedefov, 2010*; *Gershman & Fass, 2012*; *United Nations Office of Drugs and Crime, 2019*).

A number of rationales behind substitution behavior have been described. In economic theory, a "substitution effect" denotes a relationship between two commodities whose consumption is affected by alterations in price, availability, or an individual's income (*Bickel et al., 1990*; *DeGrandpre & Bickel, 1996*). When applied to drug markets, a substitution effect could explain observed variations in drug consumption patterns resulting from fluctuations in the street price of drugs (e.g., cocaine, alcohol, prescription drugs, and heroin). Such an effect influences drug consumers' preferences (*Caulkins, 2001*; *Caulkins & Reuter, 1998*; *Petry, 2001*; *Petry & Bickel, 1998*).

Legal status, price, purity, and availability were some of the considerations for substitutions among a sample of ecstasy and cocaine users who substituted their preferred drug for stimulant NPS (*Brunt et al., 2011*; *Measham et al., 2010*; *Van Hout & Brennan, 2012*). Users of the synthetic cathinone mephedrone—functional substitute of methylenedioxymethamphetamine (MDMA) (*Kapitány-Fövény et al., 2013*)—were undeterred by the illegal status of the substance in the United Kingdom (*Wood, Measham & Dargan, 2012*). Analyses of substitution patterns such as the substitution of alcohol, opioids, and prescription drugs for cannabis have shown that users prefer substitutes they perceive as having fewer side effects when self-treating a medical condition:Substitution of narcotic analgesics for cannabis has also been extensively documented among cannabis dispensary patients suffering from chronic pain. (*Lau et al., 2015*; *Lucas, 2012*; *Lucas et al., 2012*; *Lucas et al., 2016*; *Reiman, 2009*; *Subbaraman, 2014*). Substitution of cannabis for alcohol was observed when users sought a replacement substance to prevent them from relapsing to a previous drug-use during periods of abstinence (*Peters & Hughes, 2010*). Users may also substitute their usual route of administration, e.g., switching from injection to oral or smokable forms of a drug to mitigate the harms of injection drug-use (*Des Jarlais et al., 2007*).

Drug substitution with illicit or non-prescribed substances may have severe implications for the health of drug users. Acute drug related toxicities can arise when drug users substitute a familiar drug for one of higher potency. Drugs such as cannabis that are not considered highly toxic are sometimes substituted for more harmful substances such as synthetic cannabinoids (*Stevens et al., 2015*). Similarly, the substitution of MDMA with more potent and highly toxic synthetic cathinones has also been reported (*Prosser & Nelson, 2012*) resulting in hospitalizations and deaths (*Zaami et al., 2018*). Substitution may also pose a health threat in that it serves as a behavioral link between drug initiation and polydrug use patterns. This further complicates the addiction syndrome.

The "gateway theory" which describes the progressive transitioning from alcohol and tobacco, through cannabis, and onward to other psychoactive substances is a theoretical causal link between drug initiation and polydrug use (*Kandel, 1975*; *Kandel, Yamaguchi & Chen, 1992*; *Kandel & Kandel, 2015*). Polydrug use itself has been associated with mental ill-health (*Baggio et al., 2014*), psychological morbidity, suicide (*Beswick et al., 2001*; *Darke, 2004*), higher all-cause mortality, and with worse clinical outcomes (*Brecht et al., 2008*; *Darke et al., 2008*; *John, Kwiatkowski & Booth, 2001*; *Williamson et al., 2006*). Empirically, the causal links between alcohol, tobacco, cannabis, and "harder" drugs have not been robustly established and marked heterogeneity in gateway patterns have been noted across populations (*Degenhardt et al., 2010*; *Kandel, Yamaguchi & Klein, 2006*; *Mayet et al., 2016*). Together with the numerous putative demographic, substance-specific, and psychopathological risk factors for polydrug use, substitution could be considered to be a possible predictor of progression to polydrug use. Therefore, empirical characterization of a substitution phenomenon has considerable potential in research and can be used to develop informed mitigation and education strategies targeting prevention.

Cannabis figures consistently among the most prevalent substances of abuse in surveys of current drugusers (*European Monitoring Centre for Drugs and Drug Addiction, 2019*;

*Substance Abuse & Mental Health Services Administration,, 2019*). Heroin use and heroin-related deaths appear to be on the rise in the United States (*Compton, Jones & Baldwin, 2016*; *Martins et al., 2017*; *O'Donnell, Gladden & Seth, 2017*). Heroin remains the main drug-type for which people receive treatment in both Europe and Asia (*United Nations Office of Drugs and Crime, 2019*). In Israel, heroin, cannabis and cocaine are the most commonly reported drugs among treatment seeking Substance Use Disorder (SUD) individuals (*Shapira, 2019*). Cannabis use was also found to be prevalent as a primary substance among dual diagnosis patients being treated in Israel (*Rosca et al., 2018*).

In the last decade, opioids such as the fentanyl analogues have emerged in drug markets as potent and highly toxic replacements for older established opioids such as heroin and oxycodone (*Misailidi et al., 2018*). Coincidentally, NPS such as synthetic cathinones, synthetic cannabinoids, and synthetic phenethylamines have also established their presence in drug markets, thus increasing the repertoire of stimulants, hallucinogens, and sedatives available to drugusers *Helander & Bäckberg, 2017*;*Miliano et al., 2016*).

Given the high prevalence of cannabis and heroin use—and current availability of various potential substitutes for heroin and cannabis—it is important to examine which drugs are frequently used to substitute for them. Accordingly, the main purpose of this study was to determine the most frequent substitutes of heroin and cannabis among those high-risk drugusers reporting a preference for either one of these aforementioned substances. We further aimed to identify demographic, treatment, and drug-use characteristics of substituters. Finally, we described correlates for various substitution patterns among substituters of heroin or cannabis.

## MATERIALS & METHODS

Ethical approval was received from the Institutional Review Board of the Israel Ministry of Health (reference 20/2017). In this study, interviews were conducted with 592 high-risk drug users between October 2017 and October 2018 undergoing pharmacological and psycho-social treatment at one of the in-patient hospitalization facilities, dual-diagnosis treatment ambulatory units, and OMT centers (known colloquially in Israel as "Methadone Centers") under the supervision of the Ministry of Health in Israel. Recruiting used non-probabilistic quota sampling segmented to 10% females and 90% males. Segmentation by sex was done to provide a representation of the distribution of females among treatment-enrolled individuals, as reported by the Ministry of Health (*Rosca et al., 2019*). Eligible and volunteer participants were provided with written informed consent forms. The consent form was read aloud to those who preferred so.

For this research, we adopted *Reiman*'s (*2009*) definition of substitution as the use of one psychoactive substance rather than another—this may be a temporary or permanent replacement or in conjunction with another drug; and based on perceptions of cost, availability, safety, legality, and substance attributions. Our modified operational definition of a "substituter" is as follows: a druguser who declared (a) having a primary, habitual, or preferred substance (excluding tobacco); (b) that the substitute drug or drugs replaced his/her primary/habitual/preferred substance; but (c) the substitute drug was not used for
enhancement, suppression, supplementation, or mitigation of the subjective effect ("the high") of the preferred substance. Inclusion criteria included individuals who engage in high-risk drug use operationally defined by the European Monitoring Centre for Drugs and Drug Addiction as the use of psychoactive substances (excluding alcohol, tobacco and caffeine) intensively or in risky combinations and/or by high-risk routes of administration (e.g., injection) in the past 12 months (*European Monitoring Centre for Drugs and Drug Addiction, 2018*). Individuals were aged 18 years and older and currently undergoing pharmacological and/or psycho-social treatment for substance abuse. The Individuals interviewed were not at comparable stages of treatment: In-patient and dual-diagnosis enrolled individuals were at the initial stages of their treatment, receiving pharmacological and psycho-social treatment. OMT patients interviewed received ambulatory maintenance and psycho-social treatment at dedicated centers, and were at a more advanced treatment stage, having previously attended in-patient or dual-diagnosis treatment.

The interviews were conducted using a structured 54-item questionnaire containing four sections. The first section was about the basic demographic characteristics of the interviewee including age, age at onset-of-use, sex, and population group. The second section was about the past year use of specific drugs using a 22-item list of all common drugs in Israel along with their colloquial "street names", which were obtained from the Israeli Police Drug Research Section. Responders could also report a drug not mentioned in the 22-item list. The third section consisted of questions on drug preference, substitution, and motivations for substituting a drug-of-preference. This included the main question on substitution: ("Have you ever knowingly replaced or were compelled to replace your substance-of-preference for another substance?"). We also recorded the circumstances of substitution (place of substitution and substitute substance), and included questions on the lifetime frequency of substitution, the relationship between the drug-of-preference, the substance most used, and the substance most craved by the user. Additionally, we asked the responder about two separate substitution occasions that the user remembers: the substances used as substitutes and the motivations for substituting with these substances.

The population groups were analyzed considered the fact that Arab and Jewish drugusers have different drug-use trajectories and drug-preferences that can manifest themselves in diverging substitution patterns. Previous studies have found differences in severity of addiction, personality structure (*Jaraisy, 2011*) and rationality between Arab and Jewish drugusers in Israel (*Jaraisy, 2003*). Differences were also described for Arab adolescents, compared to Jewish adolescents (*Azaiza, Bar-Hamburger & Moran, 2008*). Indeed, recent population-based surveys in Israel present separate figures for these two population groups (*Harel-Fisch & Ezrachi, 2017*).

The last section contained further demographic and treatment-related questions on marital status, education, whether they were newly admitted patients, or readmitted patients. Interviews were conducted individually by research assistants trained by the main researcher and employed by the research university. All questions were read aloud to participants. No dedicated questionnaires on substitution were located from previous studies, so an unstandardized questionnaire was used with preliminary testing on 30 participants.

The reliability of the substitution measure (whether the study participant substituted his/her preferred drug) was assessed via the test/retest procedure among 30 randomly selected individuals in treatment. Cohen's $\kappa$ was run to determine the level of agreement between our substitution measure and the statement, "I always mix my preferred substance with the substance that I claimed substituted it." There was no agreement between the answers, $\kappa = -0.19$ (95% CI [0.66–0.28]; $p < .0005$) suggesting that interviewees distinguished between substitution and supplementation—the addition of an additional substance to their substance-of-preference. McNemar's test showed that the proportion of responders who answered contrarily to the main question ("have you ever substituted your substance of preference?") in the test and re-test procedure was not statistically different ($p = 0.687$).

This paper presents the secondary outcomes of a larger study that primarily examines the motivations of users to substitute their substance-of-preference. All participants of the current study were also asked questions on motivations for substitution in the same session. Hence, the results of this study are not expected to change with the conclusion of the main study.

The primary outcome measure was a report of the first and last substitutions of the participants' substance-of-preference. We described the most common substitution pattern reported for heroin or cannabis and used it as the reference category for analysis. Later, four lesser common substitutions were reported as the comparator variables. One of the comparator patterns of substitution used in this analysis, and named "all other substitutions" refers to the grouping of all other substitution patterns of lesser frequency than the other comparator patterns. Identification of demographic, treatment, and drug-use variables associated with substitution was carried-out using a $\chi^2$ test of independence. Lastly, multinomial logistic regression was performed to allow for comparisons of various potential predictors for the most commonly reported substitution patterns for heroin and for cannabis.

## RESULTS

### Sample characteristics

The median age of the overall sample was 45 years (range: 18–76), and the age at onset-of-use was 16 (range: 8–65). Lifetime substitution was reported by 448 (75.7%) of the 592 drug users interviewed of whom 253 (42.7%) reported they were only occasional substituters, i.e, having substituted their drug of-preference fewer than five occasions during their lifetime. Among those reporting lifetime substitution, 360 (80.4%) reported in detail two different substitution patterns carried-out during their lifetime, while the rest only reported one. When a broad classification of drug groups was applied (narcotics (i.e., opioids), stimulants, depressants, hallucinogens, and cannabinoids) (*Drug Enforcement Agency, 2017*), nearly two-thirds (65.1%) of the reported substitutions were made with substances from the same group. The overall sample characteristics are summarized in Table 1.

Table 2 shows that lifetime substitution was significantly associated with responders' age at onset-of-use, population group (Arab or Jewish), being in treatment for the first

**Table 1  Overall sample characteristics ($N = 592$).)**

| Variable | | Median or frequency % |
|---|---|---|
| Age, yr | | 45 (Range: 18–76) |
| Age at onset-of-use, yr | | 16 (Range: 8-65) |
| Population group | Jewish | 84.1% |
| | Arab | 15.9% |
| Sex | Female | 8.6% |
| | Male | 91.4% |
| Treatment center | Opioid Maintenance | 44.1% |
| | In-Patient | 32.4% |
| | Dual diagnosis | 23.5% |

time, and past-year use of heroin, stimulants, hallucinogens, or NPS. No significant differences were identified for age, sex, education, marital status, and past-year use of cannabis and dissociatives between substituters and non substituters. The interviews of all 448 participants who reported substitution yielded a total of 275 substitution events reported by users who reported a preference for cannabis and 351 substitution events from participants who reported a preference for heroin.

## Heroin substituters

Figure 1 summarizes the top four substitutions among heroin-preferring patients: illicit, non-prescribed ("street") methadone (35.9%), synthetic oral and transdermal opioids (17.7%), cocaine (16.5%), and cannabis (8.5%). These four reported patterns accounted for nearly 80% of all reported heroin substitution patterns. The characteristics of heroin substituters are shown in Table 3.

The analysis of substitution patterns of heroin demonstrated that substitution for illicit "street" methadone was the most common pattern reported among heroin substituters, Accordingly, in multinomial logistic regression of heroin substitution patterns, substitution for illicit "street" methadone was used as the reference category while prescription oral and transdermal opioids, cocaine, cannabis, and "all other substances" (see Fig. 2) were used as the comparator variables. Further regression analysis revealed that treatment-center type where the patient was enrolled at the time of the interview, age at onset-of-use, and education were significantly associated with some of the five analyzed substitution patterns for heroin (Table 4): The odds were 2.4 times lower (Odds Ratio [OR] = 0.41, 95% Confidence Interval [CI] = 0.20–0.58) for substitution for substances in the "others" group compared to street methadone among heroin substituters attending OMT. The odds were also 2.3-fold lower (OR = 0.44, CI [0.20–0.98]) for substitution for cocaine compared to street methadone among heroin substituters attending OMT vs. other treatment centers (in-patient, and dual diagnosis centers). Hence, attending OMT centers increased the likelihood of substitution of heroin for street methadone compared to cocaine or "all other substances".

Having a higher level of education increased the likelihood of substitution for street methadone versus substitution for substances of the "all other substitutions" group,

**Table 2  Demographic, treatment, and drug-use attributes of patients who reported substitution ($N = 448$).**

| | | | p-value[*] | Nonparametric test results[**] | |
|---|---|---|---|---|---|
| Age (years; median, range) | 44.0 | 18–76 | 0.184 | $U = 29,883, z = -1.33$ | |
| Age at onset-of-use (years; median, range) | 16.0 | 9–52 | <0.005 | $U = 25,860, z = -3.59$ | |
| | **N** | **%** | | $\chi^2$[**] | **Cramer's V** |
| Sex (% male) | 407 | 90.8 | 0.412 | 0.674 | 0.034 |
| Population group | | | 0.008 | 7.057 | 0.109 |
|   Arab | 61 | 13.6 | | | |
|   Jewish | 387 | 86.4 | | | |
| Education | | | 0.730 | 0.630 | 0.033 |
|   <High school education | 140 | 31.3 | | | |
|   High school or above | 308 | 68.7 | | | |
| Marital status | | | 0.205 | 3.170 | 0.073 |
|   Single | 220 | 49.1 | | | |
|   Married | 100 | 22.3 | | | |
| Divorced/widowed/separated | 128 | 28.6 | | | |
| Readmission to treatment | 127 | 28.3 | 0.026 | 4.962 | 0.092 |
| Past-year substance use | | | | | |
|   Stimulants[a] | 264 | 58.9 | 0.006 | 7.575 | 0.113 |
|   Cannabis | 216 | 48.2 | 0.281 | 1.164 | 0.044 |
|   Heroin | 166 | 37.1 | 0.012 | 6.240 | 0.103 |
|   Hallucinogens[b] | 74 | 16.5 | 0.015 | 5.879 | 0.015 |
|   MDMA | 85 | 19.0 | 0.017 | 5.683 | 0.098 |
|   Dissociatives[c] | 26 | 5.8 | 0.072 | 3.237 | 0.074 |
|   NPS[d] | 120 | 26.8 | <0.005 | 13.749 | 0.152 |

**Notes.**

[*]Comparisons made using Mann Whitney U tests for medians and Pearson Chi-square test for proportions.

[**]Comparison vs. non-substituters ($n = 144$).

[a]e.g., Cocaine, methamphetamine, cathinone, amphetamines, methylphenidate or drugs having similar effects.

[b]Reported as LSD, DMT, psilocybin, synthetic phenethylamines, or drugs having similar effects

[c]PCP, ketamine or similar substances

[d]Synthetic cannabinoid receptor agonists (SCRA) or synthetic cathinone

prescription opioids, or cocaine: Among heroin users reporting less than a high-school education, the odds were 2.5 times lower (OR = 0.40, CI [0.21–0.76]) for substitution for substances in the " all other substitutions " group. Similarly, the odds were 3.3 times lower (OR = 0.30, CI [0.15–0.62]) for substitution for prescription opioids, and 2.6 times lower (OR = 0.38, CI [0.18–0.77]) for substitution for cocaine compared to street methadone. Finally, the odds of substituting heroin for prescription opioids compared to street methadone decreased by a factor of 1.1 (OR = 0.90, CI [0.84–0.99]) for each increase of one year in the age at onset-of-use. Thus, an older age at onset–of–use use increased the likelihood of substituting for street methadone

## Cannabis Substituters

Among cannabis-preferring patients, substitution for SCRA (synthetic cannabinoid receptor agonists) was the most frequently reported pattern (33.5%) followed by alcohol

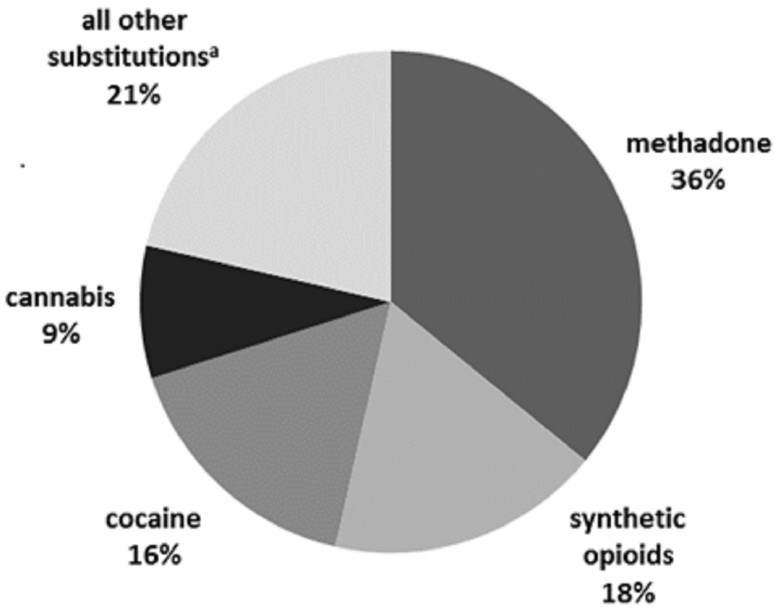

**Figure 1** **Summary of reported drug substitutions among heroin substituters** ($N = 351$). [a]LSD, lysergic acid diethylamide; Prescription opioids, e.g., oxycodone, fentanyl patches; Non-opioid analgesics, e.g., NSAIDs, dipyrone; MDMA, methylenedioxymethamphetamine Benzodiazepines, e.g., diazepam, clonazepam; Synthetic cathinones, e.g., mephedrone, ephedrone; Gabapentinoids, e.g., gabapentin, pregabalin. Note: Numbers were rounded to the next highest integer.

**Table 3** **Demographic and treatment-related characteristics of substituters of heroin** ($N = 351$).

|  |  | N | Percent |
|---|---|---|---|
| Treatment Center type | Opioid maintenance treatment | 229 | 65.2 |
|  | In-patient | 103 | 29.4 |
|  | Dual-diagnosis | 19 | 5.4 |
| Treatment Status | Readmitted patient | 78 | 22.2 |
| Sex | Female | 20 | 5.7 |
|  | Male | 331 | 94.3 |
| Population Group First time in treatment | Jewish | 297 | 84.6 |
|  | Arab | 54 | 15.4 |
| Education | <high school | 139 | 39.6 |
|  | high school/academic | 212 | 60.4 |
| Marital status | Married (currently) | 97 | 27.6 |

(16.0%), cocaine (13.5%), and cannabis resin ("hashish"), or concentrate ("dabbing") (6.9%) (Fig. 3). These four substitutions of cannabis accounted for nearly 70% of all reported substitutions for this substance. The characteristics of cannabis substituters are presented in Table 5.
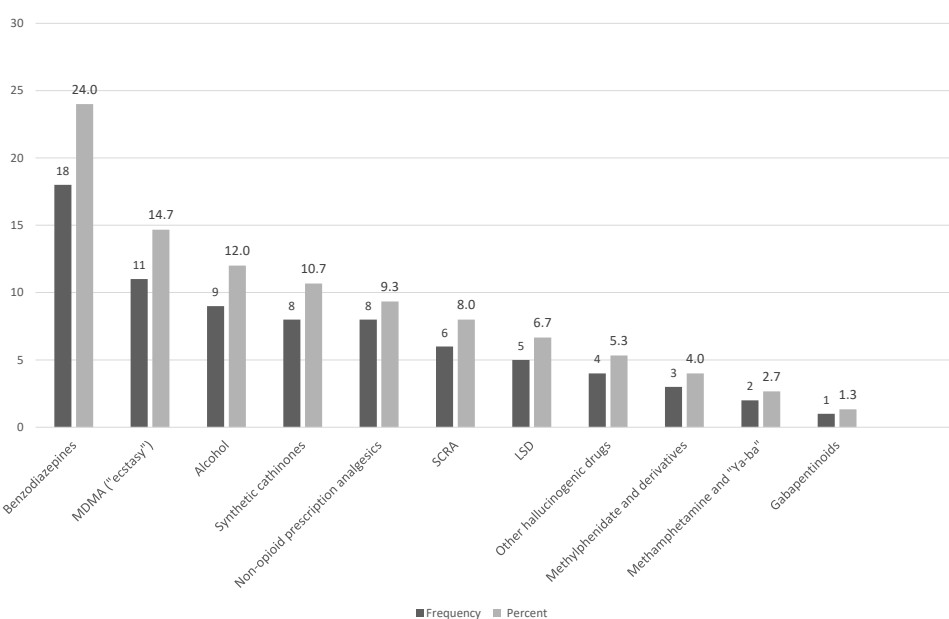

**Figure 2** **Substances included in the- "other substitutions" group, participants who substitute heroin (N = 75).** Benzodiazepines, e.g., diazepam, clonazepam. MDMA = methylenedioxy methamphetamine. Synthetic cathinones, e.g., mephedrone, ephedrone. Non-opioid analgesics, e.g., NSAIDs, dipyrone; SCRA = synthetic cannabinoid receptor agonists. LSD, lysergic acid diethylamide. Other hallucinogenic drugs, e.g., N,N-Dimethyltryptamine, 2C-B. Gabapentinoids, e.g., gabapentin, pregabalin.

**Table 4** **Demographic, and treatment related correlates for substitution pattern among heroin users, when illicit, non-prescribed methadone substitution (N = 351) is used as the reference category.**

| Substitution for: | Oral prescription opioids and transdermal fentanyl Odds Ratio (95% CI) | Cocaine | Cannabis | All other substitutions[a] |
|---|---|---|---|---|
| Parameter (Reference category) | | | | |
| Age at interview | 1.00 (0.97–1.04) | 1.01 (0.98–1.05) | 1.00 (0.95–1.05) | 0.99 (0.96–1.03) |
| Age at onset of drug use | 0.90 (0.84–0.98)[*] | 0.99 (0.93–1.05) | 0.99 (0.92–1.08) | 0.98 (0.92–1.04) |
| OMT[b] patient (ref: in-patient/dual-diagnosis) | 1.54 (0.71–3.38) | 0.44 (0.20–0.98)[*] | 1.14 (0.39–3.31) | 0.41 (0.20–0.58)[*] |
| Sex - Female (ref: male) | 1.17 (0.26–5.30) | 2.49 (0.68–9.08) | 1.38 (0.24–7.81) | 1.33 (0.33–5.31) |
| Population group - Arab (ref: Jewish) | 0.63 (0.26–1.54) | 3.62 (0.97–13.53) | 1.87 (0.47–7.49) | 1.39 (0.57–3.42) |
| First time in treatment | 0.55 (0.24–1.26) | 0.56 (0.24–1.29) | 0.81 (0.29–2.24) | 0.60 (0.28–1.29) |
| <High school education | 0.30 (0.15–0.62)[*] | 0.38 (0.18–0.77)[*] | 0.54 (0.23–1.26) | 0.40 (0.21–0.76)[*] |
| Marital status –not married[c] (ref: married) | 1.01 (0.45–2.25) | 0.91 (0.41–1.99) | 0.94 (0.36–2.49) | 0.87 (0.42–1.78) |

**Notes.**
[*]$p \leq 0.05$.
[a]See Fig. 2 for substitution frequencies.
[b]Opioid Maintenance Treatment.
[c]Single/divorced/widowed/separated.

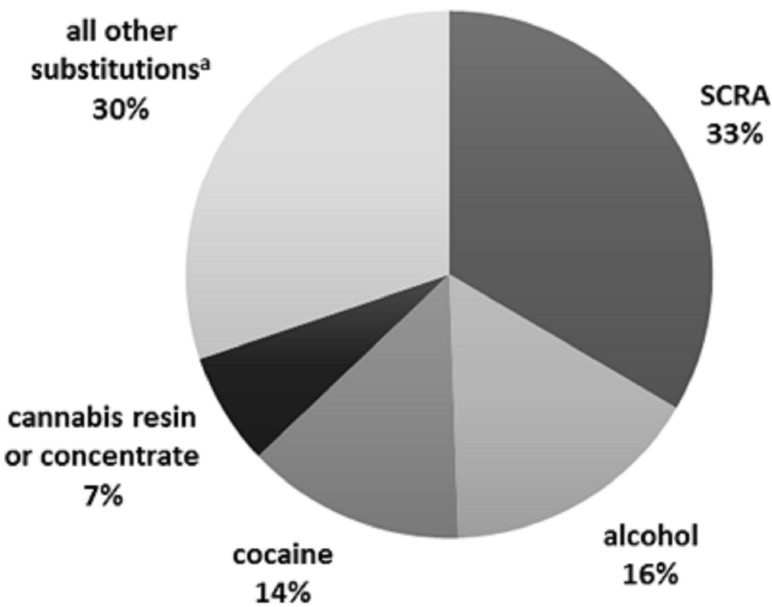

**Figure 3** Summary of reported drug substitutions among cannabis substituters (*N* = 275). [a]LSD, lysergic acid diethylamide; Prescription opioids, e.g., oxycodone, fentanyl patches; Non-opioid analgesics, e.g., NSAIDs, dipyrone; MDMA, methylenedioxymethamphetamine Benzodiazepines, e.g., diazepam, clonazepam; Synthetic cathinones, e.g., mephedrone, ephedrone; Gabapentinoids, e.g., gabapentin, pregabalin. Note: Numbers were rounded to the next highest integer.

**Table 5 Demographic and treatment treatment-related characteristics of substituters of cannabis (*N* = 275).**

|  |  | N | Percent |
|---|---|---|---|
| Treatment Center type | Opioid maintenance treatment | 55 | 20.0 |
|  | In-patient | 103 | 37.5 |
|  | Dual-diagnosis | 117 | 42.5 |
| Treatment Status | Readmitted patient | 102 | 37.1 |
| Sex | Female | 25 | 9.1 |
|  | Male | 250 | 90.9 |
| Population Group First | Jewish | 230 | 83.6 |
|  | Arab | 45 | 16.4 |
| Education | <high school | 73 | 26.5 |
|  | high school/academic | 202 | 73.5 |
| Marital status | Married (currently) | 60 | 21.8 |

Multinomial logistic regression analysis of cannabis substitution patterns used substitution for SCRA as the reference category while the following patterns: alcohol, cocaine, and cannabis resin/concentrate, and "all other substances" (see Fig. 4) were used as the comparator variables.

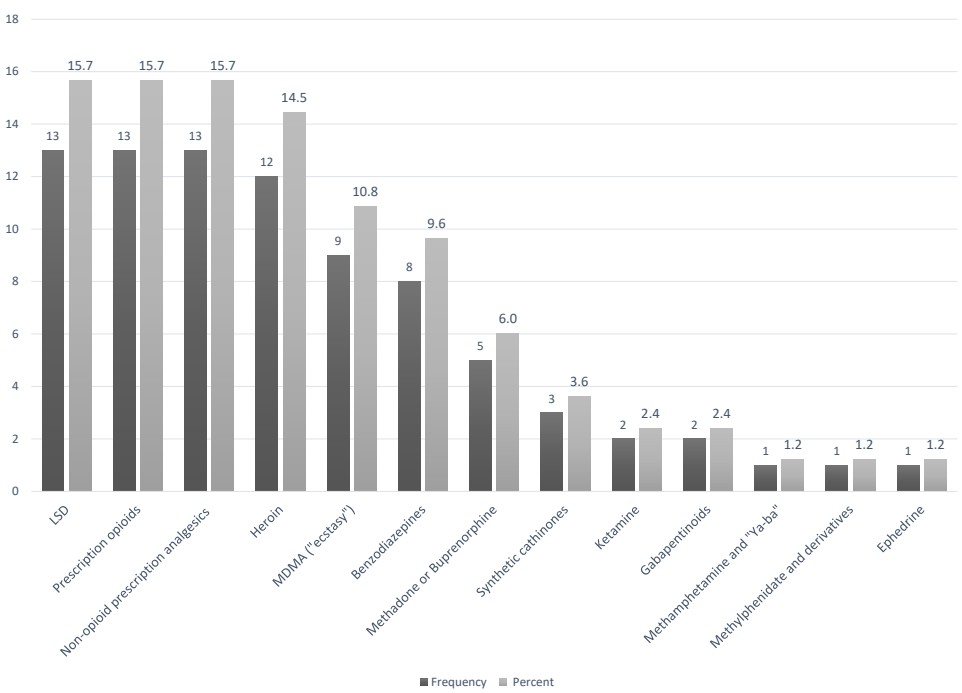

**Figure 4    Substances included in the "other substitutions" group, participants who substitute cannabis ($N = 83$).** LSD, lysergic acid diethylamide. Prescription opioids, e.g., oxycodone, fentanyl patches. Non-opioid analgesics, e.g., NSAIDs, dipyroneMDMA = methylenedioxymethamphetamine. Benzodiazepines, e.g., diazepam, clonazepam. Synthetic cathinones, e.g., mephedrone, ephedrone. Gabapentinoids, e.g., gabapentin, pregabalin.

Table 6 shows that among cannabis substituters, age, treatment center type, readmission, education, and population group were significantly associated with substitution pattern. The odds of substituting cannabis for substances of the "all other substitutions" group were 1.05 times higher compared to SCRA (OR = 1.05, CI [1.02–1.08]) for each increase of one year in the age of cannabis users. It was 5.9 (OR = 5.94, CI [1.92–18.35]) times higher for Arab cannabis users compared with Jewish cannabis users. Conversely, the odds of substituting cannabis for substances of the "all other substitutions" group were 6.3 times lower than those of SCRA (OR = 0.16, CI [0.08–0.36]) for cannabis substituters attending dedicated dual-diagnosis centers vs. other treatment centers (in-patient and OMT). This was 3.3-fold higher (OR = 3.33, CI [1.56–7.10]) for cannabis substituters attending treatment for the first time.

The odds were 3.0 times lower (OR = 0.33 CI [0.13–0.83]) for substitution for cocaine compared to SCRA among cannabis substituters attending dedicated dual-diagnosis centers vs. other treatment centers (in-patient, and OMT centers) and 1.12 times lower (OR = 0.89, CI = 0.80–0.99) for each increase of one year in the age at onset-of-use. Conversely, the odds of substituting cannabis for cocaine, compared to SCRA, were 7.1 (OR = 7.12, CI [1.75–28.9]) times higher for Arab cannabis users compared with Jewish cannabis users. Finally, the odds were 6.7 times lower (OR = 0.15, CI [0.03–0.80]) for substitution of

**Table 6  Demographic, and treatment-related correlates for substitution pattern among cannabis users, when substitution for synthetic cannabinoids (SCRA) (N = 275) is used as the reference category.**

| Substitution for: | Alcohol | Cocaine | Cannabis resin ("hashish") or extract ("dabbing") | All other substitutions[a] |
|---|---|---|---|---|
| | | Odds Ratio (95% CI) | | |
| **Parameter (Reference category)** | | | | |
| Age at interview | 1.00 (0.96–1.04) | 1.02 (0.98–1.06) | 1.02 (0.97–1.07) | 1.05 (1.02–1.08)[*] |
| Age at onset of drug use | 0.98 (0.92–1.05) | 0.89 (0.80–0.99)[*] | 0.98 (0.88–1.09) | 1.01 (0.95–1.07) |
| Dual diagnosis patient (ref: in-patient/OMT[b]) | 1.03 (0.43–2.45) | 0.33 (0.13–0.83)[*] | 0.39 (0.12–1.24) | 0.16 (0.08–0.36)[*] |
| Sex - Female (ref: male) | 2.18 (0.62–7.63) | 0.75 (0.13–4.35) | 0.67 (0.07–6.65) | 1.12 (0.33–3.83) |
| Population group - Arab (ref: Jewish) | 3.85 (0.90–16.58) | 7.12 (1.75–28.9)[*] | 0.41 (0.10–1.66) | 5.94 (1.92–18.35)[*] |
| First time in treatment | 1.25 (0.56–2.85) | 0.74 (0.31–1.77) | 1.84 (0.62–5.44) | 3.33 (1.56–7.10)[*] |
| <High school education | 0.65 (0.24–1.75) | 0.93 (0.33–2.66) | 0.15 (0.03–0.80)[*] | 0.97 (0.43–2.17) |
| Marital status –Not married[c] (ref: married) | 0.74 (0.24–2.27) | 0.40 (0.13–1.21) | 3.02 (0.54–16.79) | 1.04 (0.40–2.70) |

**Notes.**

[*]$p \leq 0.05$.

[a]See Fig. 2 for substitution frequencies.

[b]Opioid Maintenance Treatment.

cannabis resin or concentrate versus SCRA among herbal cannabis substituters having an education level below high-school.

## DISCUSSION

In this sample, more than three-quarters of treatment-enrolled drug dependent individuals reported having substituted their preferred drug making substitution the rule rather than the exception. In our analysis of substituters, past-year NPS use was significantly associated with substitution. This result is consistent with the notions of NPS appeal to drug users as substitution drugs for older established substances (*Brennan & Van Hout, 2012*; *Measham et al., 2010*; *Van Hout & Brennan, 2012*) rather than of NPS as supplements, which do not displace drug repertoires (*Moore et al., 2013*; *Van Amsterdam et al., 2015*). Hence, it was expected that NPS would play a prominent role as substitutions for cannabis and heroin. However, besides the use of SCRA, there was no evidence that other NPS served as frequent and important substitutes for heroin and cannabis.

The notion of pharmacological similarity as the basis of substitution potential is supported by numerous pre-clinical drug discrimination studies (*Solinas et al., 2006*; *Stolerman, 2014*). However, within our samples of cannabis and heroin substituters, only 40% of cannabis substitution (for SCRA and "Hashish") and just over 50% of all heroin substitutions (for street methadone and synthetic opioids) were carried out within the same broad drug-class group as the preferred substance. Hence, in our sample, the substitutes did not necessarily share similar subjective or physiological effects with the drug they replaced. Nevertheless, findings show that the most commonly reported substitutions were for drugs within the same drug-class group and with similar effects e.g., (heroin for methadone and cannabis for SCRA) reflecting the importance of users' preference for a specific effect.

Substitution for prescription opioids was the second most frequent substitution reported among heroin substituters. Prescription opioids are commonly diverted from licit sources and used by primary heroin users (*Cicero, Ellis & Kasper, 2017*; *Compton, Jones & Baldwin, 2016*; *Davis & Johnson, 2007*). In this study, the patients did not report substitution with injectable or oral fentanyl. The use of potent fentanyl analogues by heroin users has been associated with many deaths and hospitalizations in North America (*Ciccarone, 2017*; *Pardo et al., 2019*). Nonetheless, fentanyl patches have become increasingly available to heroin-users in the illicit drug market and have been implicated in at least three deaths in Israel (*Herman, 2019*). Hence, the mounting evidence of substitution for prescription opioids among Israeli heroin-users is of concern.

Substitution for cocaine was the third most prevalent pattern reported among heroin substituters. Similar substitutions were reported in earlier studies in which heroin unavailability was associated with increased consumption of cocaine (*Degenhardt et al., 2005*; *Degenhardt et al., 2010*). Thus, this study provides further evidence of the displacement of heroin by cocaine among some heroin users despite both drugs having dissimilar effects.

The study described higher odds for substituting cannabis for cocaine and cannabis substitution for ''all other substitutions'' among Arab cannabis substituters versus Jewish cannabis substituters. Jewish cannabis substituters had higher odds for SCRA substitution. This is consistent with earlier surveys, which had prevalence figures for NPS use that is significantly higher among the Jewish population (*Harel-Fisch & Ezrachi, 2017*). Additionally, a higher age at onset of drug-use was associated with lower odds for cocaine substitution versus SCRA substitution. In Israel, the mean age of onset of use for NPS among the general adult population (18.5) is considerably lower than that for opioid use (*Harel-Fisch & Ezrachi, 2017*; *Inbar, 2015*). Thus, the appeal of SCRA at a younger age could imply that the easier access and lower street price of these NPS is particularly attractive to younger drugusers.

The high frequency of reports of illicit ''street'' methadone as a substitute for heroin is puzzling. Moreover, the results demonstrate that being enrolled in OMT was significantly associated with substitution for street methadone. Trade in methadone among OMT patients to supplement income (*Johnson & Richert, 2015*; *Lauzon et al., 1994*) or acquire funds to buy other drugs has been documented in some studies of OMT patients (*Fountain et al., 2000*). This premise could explain how some of the illicit methadone is made available to patients outside treatment. Some hypotheses could be proposed for this common pattern of heroin substitution for illicit methadone. The pattern could be partly explained by the need of patients to manage withdrawal symptoms. The need for OMT patients to resort to substituting with street methadone could also be evidence that current treatment does not satisfy the users' needs or that methadone treatment is lacking in dose and/or availability (*Roche, McCabe & Smyth, 2008*). Methadone dose is indeed, an important predictor of success of, and continued compliance to OMT (*Strain, 2006*). Furthermore, most guidelines require methadone doses to be tailored using clinical discretion because some patients require higher doses than those usually indicated for OMT (*Baxter et al., 2013*). Ministry of Health guidelines allow for administrating higher doses (usually above

120 mg) if withdrawal symptoms are still apparent (Department for the Treatment of Substance Abuse, 2014), but concerns over diversion, overdose, and toxicity might have induced clinicians to under-dose (*Duffy & Baldwin, 2012*; *Lin & Detels, 2011*).

Inaccessibility may force heroin-dependent patients who require prescription methadone, but are unable to consistently visit these centers, to purchase it illicitly (*Carroll, Rich & Green, 2018*). In Israel, OMT and in-patient centers are usually located in industrial areas in the periphery of cities with limited hours of operation. They are not easily accessible by public transport. This is exacerbated by opposition from residents and business owners against the presence of drug-treatment centers near residential and commercial areas (known colloquially as NIMBY—Not in My Back Yard; *Bernstein & Bennett, 2013*; *Tempalski et al., 2007*) a phenomenon also reported in Israel (*Time out, 2019*). The Ministry of Health in Israel has indeed affirmed that the present geographic distribution of OMT and in-patient units remains insufficient despite significant investment in new drug treatment centers (*Rosca et al., 2019*). Moreover, within OMT centers, one-third of treated patients were found to be active users of street drugs (*Rosca et al., 2018*; *Rosca et al., 2019*). The figures may be higher considering under-reporting and the use of substances not detectable by current kits. The Israeli public drug-treatment system uses a mix of sanctions and incentives when patients fail to comply with center regulations requiring abstinence from street drug use. Sanctions range from simple warnings to the transfer of a patient to another facility (*Department for the Treatment of Substance Abuse, 2014*). Incentives include a partial waiver for over 60% of the treatment bill and providing the patient with take-home doses of methadone thus avoiding the compulsory daily attendance requirements in some centers (*Department for the Treatment of Substance Abuse, 2014*; *Israel Ministry of Health, 2018*). Hence, another reason for the frequent substitution of heroin for street methadone may be that its use does not incriminate users when it is detected by current urine screening methods employed by centers because it is routinely prescribed to users. Accordingly, heroin-dependent patients may be able to access higher drug doses of methadone in the streets without being incriminated during treatment. Further inquiry into the motivations for substitution for street methadone among drugusers could shed light on this phenomenon.

SCRA was the most common substitution substance reported among cannabis substituters. In the past, the appearance of SCRA—as legal alternatives to cannabis—in "head shops" and kiosks contributed significantly to the rise of the NPS phenomenon (*Gunderson et al., 2014*; *Vandrey et al., 2012*; *Winstock & Barratt, 2013*). SCRA are highly potent and toxic drugs and are associated with severe health consequences including psychosis (*Fattore, 2016*; *Tait et al., 2016*). In this study, dual-diagnosis patients using cannabis had comparatively higher odds of substituting it for SCRA versus other substances. Previous research described frequent SCRA use among dual diagnosis patients (*Bassir Nia et al., 2016*). Another study demonstrated a high prevalence of concurrent SCRA and cannabis use among Israeli psychiatric patients (*Shalit et al., 2016*). Concurrent use could be partially attributed to mutual substitution between these substances. In general, dual-diagnosis has been associated with increased exposure to drugs, disinhibition, and drug experimentation (*Kessler, 2004*). This factor could favor the use of NPS and other

novel substances. Cannabis use is also a form of symptom management among dual-diagnosis patients suffering from depression and anxiety (*Santucci, 2012*). It is possible that substitution of cannabis for SCRA among dual-diagnosis patients provides similar relief.

Alcohol was the second most frequent substitution substance reported among cannabis substituters. Past studies have documented that users increased their alcohol consumption in times of abstinence from cannabis (*Peters & Hughes, 2010*). Hence, the substitution of cannabis for alcohol provides further evidence that both substances affect each other's consumption through mutual replacement (*Looby & Earleywine, 2007*).

### Limitations

Although measures, such as test-retest procedures and corrections to the questionnaire were performed to provide a reliable tool for identifying substitution, subject recall bias regarding patterns of drug-use may have produced inaccurate reports of drug-use behavior (*Anthony, Neumark & Van Etten, 2009*). Second, because multinomial logistic regression analysis required combining all less-frequent substitution patterns of heroin and cannabis in the "all other substances" groups, this study cannot provide detailed analysis of these substitutions. Individually, the substitution patterns included in these groups were reported with limited frequency. Nevertheless, when grouped, these patterns comprised a significant proportion of all substitutions reported (up to 30%).

Some substances like SCRA and street methadone have distinctive appearances and presentations. Street methadone in Israel is usually diverted from OMTs and thus commonly appears in liquid form when mixed with concentrated strawberry or raspberry juice. SCRA is sold as a smokable herbal substances in small silver colored bags and marketed under the names "Nice Guy", and "Mabsuton." Nevertheless, some substances cannot be easily identified with exactitude by drug users such as some hallucinogens or synthetic stimulants that appear as a plethora of compounds in drug markets. This limits the reliability and validity of participants' reports.

Finally, this study employed a non-probabilistic quota sampling method. The choice of this sampling method was due to the difficulty of recruiting hard-to-access populations to report on the consumption of illicit substances–an act that is criminalized in most jurisdictions. A larger sample could aid in the analysis of the substitution patterns of these groups.

## CONCLUSIONS

This research demonstrates that substitution is not an unusual pattern of drug-use behavior. Substitution analysis is a promising and revealing avenue of research for understanding the behavior of drugusers and the effect of treatment policy on user behavior. Separate conclusions could be inferred from both analyses of cannabis and heroin substitution. Concerning heroin, current patterns demonstrating prominence of illicit substitution for street methadone—particularly among OMT patients—are evidence of ongoing diversion, and possible deficiencies in treatment. Second, the continuous availability of SCRA as a highly toxic alternative to cannabis must be recognized as a significant phenomenon affecting dual-diagnosis patients. More efforts should be invested in monitoring and

mitigating the use of SCRA among these patients. Overall, the phenomenon of substitution for illicit "street" substances should be further examined, as it could have detrimental effects on users' health due to the low quality of street drugs and because of potential high dose-to-dose variability.

## ACKNOWLEDGEMENTS

The author/s acknowledge the valuable administrative assistance provided by the staff at the Beer-Sheva Methadone center, Tel-Aviv Zur Aviv, Jerusalem methadone center, Maor Ashdod, and MAAMATZ Haifa centers.

### Funding

This work was supported by the Israeli Anti-Drug and Alcohol Authority (IADA) under grant 0394937. The funders had no role in study design, data collection and analysis, decision to publish, or preparation of the manuscript.

### Grant Disclosures

The following grant information was disclosed by the authors:
Israeli Anti-Drug and Alcohol Authority (IADA): 0394937.

### Competing Interests

The authors declare there are no competing interests.

### Author Contributions

- Barak Shapira conceived and designed the experiments, performed the experiments, analyzed the data, prepared figures and/or tables, and approved the final draft.
- Paola Rosca and Ronny Berkovitz conceived and designed the experiments, authored or reviewed drafts of the paper, and approved the final draft.
- Igor Gorjaltsan conceived and designed the experiments, authored or reviewed drafts of the paper, helped in recruiting patients and in questionnaire building, and approved the final draft.
- Yehuda Neumark conceived and designed the experiments, analyzed the data, authored or reviewed drafts of the paper, and approved the final draft.

### Ethics

The following information was supplied relating to ethical approvals (i.e., approving body and any reference numbers):

Institutional Review Board of the Israel Ministry of Health Institutional Review Board of the Israel Ministry of Health approved the study (reference 20/2017).

## Data Availability

A SPSS Data file for all 996 substitutions, as reported by 592 SUD treatment-enrolled individuals in Israel, is available at Zenodo: Barak Shapira. (2020). What Does Drug Substitution Behavior Tell Us About the Deficiencies in Our Drug Treatment Services?: Illicit drug-substitution behavior in a sample of high-risk drug users in Israel (Version 4.0). http://doi.org/10.5281/zenodo.3628740.

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
