# Peer review of "The switch from one substance-of-abuse to another: illicit drug substitution behaviors in a sample of high-risk drug users"

_PeerJ, doi:10.7717/peerj.9461_

## Round 0.1 · original submission · Major Revisions

This study is a snapshot of a population of drug users who interact with a drug treatment service. The reviewer of this study expresses significant concerns about the description of the methods used in the study, and the characteristics of the people surveyed. As indicated, it would be expected that enough detail is provided to enable replication of the study either locally or in a cohort elsewhere - or at least to be able to provide enough information that meaningful comparisons could be made. How were interviews structured, what were the exact questions asked, who asked them and in what context ? This is part of a larger study - what is that, and how could the conduct of that study affect the data presented here ?

While demographic data such as age and gender are standard, whether a subject is Jewish or Arab does not hold special meaning for most researchers, and indeed without a clear definition of what this means, how it was determined, and why it is important, it seems a jarring inclusion.

Finally, while drug users are very aware of what they are taking, there is probably no way for them to know exactly what it is when sourced unofficially - how reliable are self reports of "street methadone" use versus a different synthetic opioid, or identification of an SCRA ? Is it the intention to substitute that you are actually recording, rather what was actually done ?

This work is interesting, and potentially suitable for publication in PeerJ, however, it will certainly be re-reviewed if you choose to resubmit.

Reviewer 1 ·

Basic reporting

The language becomes casual in parts and there are a number of erroes in punctuation and sentence structure.
There was sufficient background and context.
The article could be restructured to flow more easily. Some of the tables could be substitued with graphs which might make it easire to understand.

Experimental design

The research question is not well defined.The methods are not described with sufficient detail to be confident that the investigation has been performed with a high technical standard, or indeed that it could be replicated.

Validity of the findings

There is not enough data in the methods on participant recruitment and what was assessed (frequency of use, period over which this was discussed) to fully appreciate the results.

Additional comments

What does drug substitution behaviour tell us about the deficiencies in our drug treatment service. Illicit drug substitution behaviours in a sample of high-risk drug users.
The title is misleading. This study describes substance use in a cohort engaged in various treatments for SUD. I don't think it can really identify deficiencies in drug treatment services.
The paragraph “Purpose of the study” is a bit confusing. The difference between aims and objectives is not helpful. The sentence “Data provided by treated individuals on current and past drug use was used to ascertain the prevalence of substitutions” might be better placed in the methods section. It is perhaps more typical to describe aims then clarify the primary and secondary outcomes n the methods section.
There are a number of problems with the methods. This current study is reported to be part of a larger study. This is not explained in sufficient detail, or indeed why the current study is being presented. The date range of the study is not provided.

The nature of the population examined is a bit unclear. There needs to be greater detail in the definition of a high risk drug user, rather than just the citation.
The paper could be improved by a number of clarifications about the population under study.
(1) The source of participants could be clarified: Detoxification and Opioid Agonist Treatment (OAT) are not the same. Line 177 describes physical detoxification in hospital, ambulatory units or in methadone centres where participants are receiving OAT. Line 194 talk about drug users undergoing detoxification. Are participants being “detoxed” in methadone centres ? The term “Physical detoxicfication” is unfamiliar and requires clarification.
(2) The stage at treatment at which participants are recruited should be clarified.
(3) A description of how participants were recruited to describe potential sources of bias from recruitment strategies would be useful. This would help identify to what extent the participant population reflects the populations attending these treatment centres.
(4) There needs to be greater detail of the questions asked of the participants. Was a standardised questionnaire used. Over what period of time prior to interview was drug use ascertained? Is recall bias therefore a problem? Is there any identification of frequency of use? Perhaps a sample questionnaire could be appended to the paper.
(5) Were the doses of OAT identified as inadequate OAT dose is a significant contributor to continuing opioid use.
The results are quite difficult to read and a bit disjointed. The order of presentation is a bit unusual. It is conventional to describe the demographic characteristics of the sample, and analysis done of that group, and then present the subgroup description and analysis sequentially. Presenting the substitution data graphically (pie chart) might be visually more appealing than a table, and easier to understand.
There are a number of typographical and grammatical errors throughout the manuscript, and the language can become a bit colloquial (eg NIMBY).

---

## Round 0.2 · accepted · Accept

Thank you for constructively responding to the issues raised by the reviewers.